# Fluctuations in influenza-like illness epidemics and suicide mortality: A time-series regression of 13-year mortality data in South Korea

Sun Jae Jung[1,2], Sung-Shil Lim[3], Jin-Ha Yoon[1,3]*

**1** Department of Preventive Medicine, Yonsei University College of Medicine, Seoul, Republic of Korea, **2** Department of Epidemiology, Harvard T. H. Chan School of Public Health, Boston, Massachusetts, United States of America, **3** The Institute for Occupational Health, Yonsei University College of Medicine, Seoul, Republic of Korea

* flyinyou@gmail.com

Fluctuations in influenza-like illness epidemics and
suicide mortality: A time-series regression of 13-
year mortality data in South Korea. PLoS ONE
16(2): e0244596. https://doi.org/10.1371/journal.
pone.0244596

School of Medicine, UNITED STATES

**Data Availability Statement:** Data are available
from the Korea statistical information system
(KOSIS) Statistics DB Internet (http://kosis.kr/eng).
The detail step is following: 1) mdis.kostat.go.kr, 2)

## Abstract

### Aims

We explored the association between influenza epidemic and suicide mortality rates in a large population using a time-series regression of 13-year mortality data in South Korea.

### Methods

Weekly suicide mortalities and influenza-like illness (ILI) were analyzed using time series regression. Regression coefficient for suicide mortality based on percentage change of ILI was calculated using a quasi-Poisson regression. Non-linear distributed lag models with quadratic function up to 24 weeks were constructed.

### Results

The association between ILI and suicide mortality increased significantly up to 8 weeks post-influenza diagnosis. A significant positive association between ILI and suicide mortality was observed from 2009, when a novel influenza A(H1N1)pdm09 virus provoked a worldwide pandemic. No meaningful association between these factors was observed before 2009.

### Conclusion

There was a significant positive relationship between ILI and suicide mortality after 2009, when a novel influenza A(H1N1)pdm09 virus provoked a worldwide pandemic.

## Introduction

In 2009, the U.S. Centers for Disease Control and Prevention reported four cases of previously healthy children who developed neurological complications after being infected with

Sign Up, 3) go to death record data under public health sector, 4) choose period of data set, 5) and download. Influenza monitoring data from 2004 to 2018 was available from KCDC website without any requirement for permission (http://www.cdc.go.kr/npt/biz/npp/iss/influenzaStatisticsMain.do).

**Funding:** This study was supported by the Basic Science Research Program through the National Research Foundation of Korea (NRF) and funded by the Ministry of Science and ICT (NRF-2019R1A4A1028155) (to SJJ). This work was supported by Korea Health Industry Development Institute through "Social and Environmental Risk Research" funded by Ministry of Health & Welfare (HI19C0052) (to JHY). The funder had no role in study design, data collection and analysis, decision to publish, or preparation of the manuscript.

**Competing interests:** The authors have declared that no competing interests exist.

pandemic influenza A (H1N1) [1]. Similar case reports from other countries have raised concerns regarding the association between influenza infection and neuropsychiatric symptoms such as depression, anxiety, changes in mental status, and seizures [2]. However, the results were conflicting, and a systematic review [3] found that the available evidence was insufficient to guide a definite conclusion regarding the association between influenza infection and neuropsychiatric symptoms. Especially in South Korea, one study dealt with the adverse psychiatric events after the treatment of H1N1, oseltamivir [4], but evidence lacked for the association between infectious epidemic itself and neuropsychiatric symptoms. In addition, a new virus subtype labeled novel influenza A(H1N1)pdm09 first emerged in 2009, even though the influenza subtype of H1N1 existed before 2009.

In population studies, the total numbers of suicide mortality have shown seasonal peaks especially during the early spring, which usually falls in February to March in Korea. These peaks overlap with seasonal peaks in epidemics of upper respiratory tract infection, which may affect the nose, throat, sinus, and larynx. In a study conducted in South Korean children with lower respiratory infections who were aged less than 18 years, two distinct peaks in viral upper respiratory tract infection was noted yearly, each during April-May and October-November [5]. The development of prevention strategies relies on an understanding of the cause of suicide mortality at a population level. However, little available evidence has supported the association between a new influenza (H1N1) virus pandemic, novel influenza A(H1N1)pdm09, and suicide in a substantial number of people, with respect to periodic changes. Therefore, we explored the association between influenza-like illness(ILI) infection and suicide mortality rates across a large population using time-series regression.

## Methods

### Ethical approval and consent to participate

The subject information was blinded prior to analysis. The Institutional Review Board (IRB) of the Yonsei University Health System approved the current study design (IRB number: Y-2017-0100).

### Study population

We obtained data for this study from the Korea Centers for Disease Control and Prevention (KCDC). As the information regarding virus genotyping was unavailable, we could not distinguish between H1N1 infection and other flu-like illness, however, we obtained information about ILI outpatient visits (per 1,000 people) weekly ranging from the 35th International Organization for Standardization (ISO) week in 2004 to the 52th ISO week in 2017. The Korean government made surveillance system for ILI at 53 hospitals, using geographic stratification. By law [6], every physician are required to report the number of patients who are suspected to suffer from ILI. The government also issued a guideline on how to define ILI for proper reporting (see S1 Box below). In this guideline, physicians should report patients with clinical symptoms with one or more virus-specific biomarkers, including influenza virus antibody or viral gene, from specimen as ILI cases.

Suicide mortality data during the same period were acquired from the national death records of the Korea National Statistical Office. This record covers all deaths throughout South Korea and classifies the causes based on the International Classification of Diseases, 10th Edition (ICD-10). To capture suicide along with intentional self-harm and sequelae of intentional self-harm, we selected X60-X84 and Y870 codes for the outcome. We included the entire population in the final analysis. For the control analysis, we obtained cancer mortality from the

same data to demonstrate that non-suicide death was not related to ILI. Death due to cancer was coded as C00~C99 according to the ICD-10 code.

## Statistical analysis

We used the weekly percent of ILI and weekly number of suicide. As the number of weekly suicide mortality showed seasonality, we fitted linear function with sine and cosine formulae [7]. We used 52.18 (365.25/7) for cyclic frequency, as not every year had 52 weeks. Generalized additive model (GAM) was used for cubic spline of seasonality, and minimal values of Akaike Information Criterion (AIC) or Bayesian information criterion (BIC) were used in the model fitting. The formula for time series analysis with GAM, as shown below.

$$g(suicide_t) = \beta_0 + f_1(ILI_{t-l}) + f_2(Season_t) + f_3(Week_t) + \varepsilon(t)$$

where: $f_s(x_t) = \sum_{i=1}^{k} b_i(x_t)\beta_i$, $\beta_i$ is the weight value applied to the $i$th basis function ($b_i$), so the $f_s(x_t)$ is smooth function of covariates $x_t$. Suicide($t$) is weekly($t$) number of suicides, and ILI($t$-$l$) is corresponding weekly ILI with lag time 0 to 8 weeks. Season and long-term trend by week are confounding factors. Every step of model building were simulated by degree of freedom (df) from 2 to 200 in the generalized linear model or GAM, and graphical approach was used to detect the optimal df of GAM spline parameter for minimal AIC or BIC. If there was no change point of AIC until a degree of freedom of 200 simulations, then BIC was used. The degree of freedom of GAM spline parameter was 5 for seasonality and 39 for other remnant trends. Then, after controlled seasonality, The partial autocorrelation plot of residuals showed no cyclic pattern.

The same simulation was used to select a non-linear distributed lag model from 0 to 24 weeks with a quadratic function. Rate ratios (RRs) for suicide mortality with 95% confidence intervals (95% CIs) based on %ILI increments were calculated using a quasi-Poisson regression to address the problem of over-dispersion. Analyses for the effects of swine flu pandemics were stratified as before and during/after 2009. For the control analysis, all processes were replicated with cancer mortality as the outcome. Data from August 2004 to December 2017 were used analysis. All statistical analyses and graphics were generated using R program, ver. 3.5 (R Foundation for Statistical Computing, Vienna, Austria).

## Results

A total of 179,839 suicide mortalities were recorded during the study period (average number of annual Korean population was 49,093,917). In terms of annual suicide deaths, 12,043 cases were reported in 2005, followed by a peak of 15,914 cases in 2011 and a gradual decrease to 12,429 cases in 2017, with seasonal fluctuations (Fig 1). There were monthly or seasonal variations in the number of suicide deaths, with peaks occurring from March to May (Fig 2). The median (interquartile range) number of weekly suicide was 278 (250–312) in March, 295 (267–320) in April, and 286 (259–321) in May. However, we could not find a significant difference in the magnitude of the association between ILI and suicide mortality across every month by linear regression.

A statistically significant increment in ILI rate was observed starting in 2009, in a time series regression analysis. The relationship between the ILI number per 1,000 people and suicide deaths exhibited a significant positive association since 2009 ($\beta$ = 0.013, $P$ = 0.018), but not before 2009 ($\beta$ = -0.066, $P$ = 0.214) (Fig 1).

The RRs (95% CIs, p value) for suicide mortality based on ILIs were as follows: 1.007 (1.003–1.012, <0.001) with no lag, 1.007 (1.003–1.011, <0.001) with a 1-week lag, 1.006 (1.003–1.010, 0.001) with a 2-week lag, 1.006 (1.002–1.009, 0.001) with a 3-week lag, 1.005

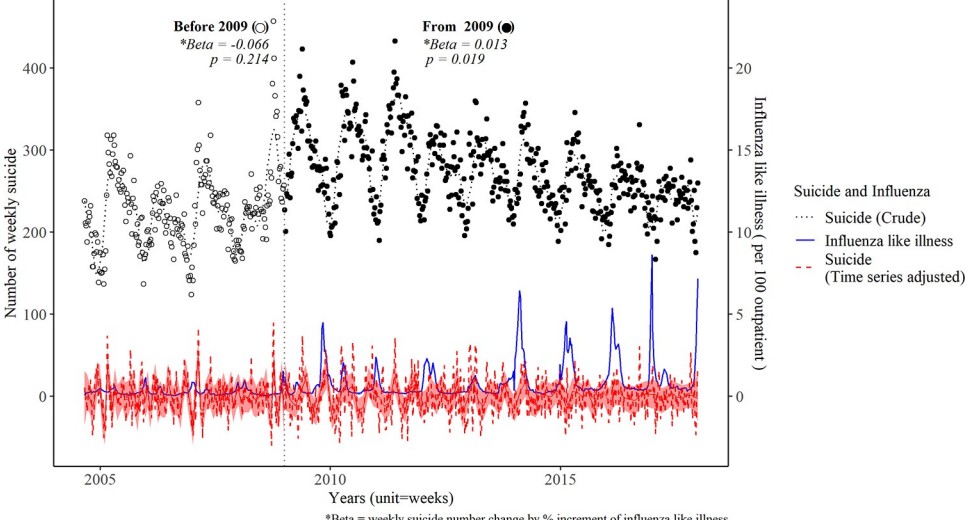

**Fig 1. Weekly numbers of suicides and influenza-like illnesses per 100 outpatients from 2004 to 2017.** *β (Beta)* indicates the change in weekly suicide numbers, considering the % increment in influenza-like illness cases; red dot line show weekly number of suicide after adjusting time series factors, and red shadow indicates 95% confidence interval of the estimated weekly suicide rate, which was assessed using a generalized additive model; blue line show influenza like illness per 100 outpatients without lag time, and horizontal line indicates the year 2009.

(1.001–1.008, 0.003) with a 4-week lag, 1.004 (1.001–1.007, 0.006) with a 5-week lag, 1.004 (1.001–1.007, 0.011) with a 6-week lag, 1.004 (1.001–1.007, 0.020) with a 7-week lag, and 1.003 (1.000–1.007, 0.033) with an 8-week lag, respectively (Fig 3). From week 9, the RRs of suicide mortality were attenuated, and the relationship of ILI with suicide death became insignificant (p values > 0.05). In the control analysis, when comparing cancer mortality with suicidal mortality, we could not find any significant relationship between influenza symptoms and cancer mortality (data not shown).

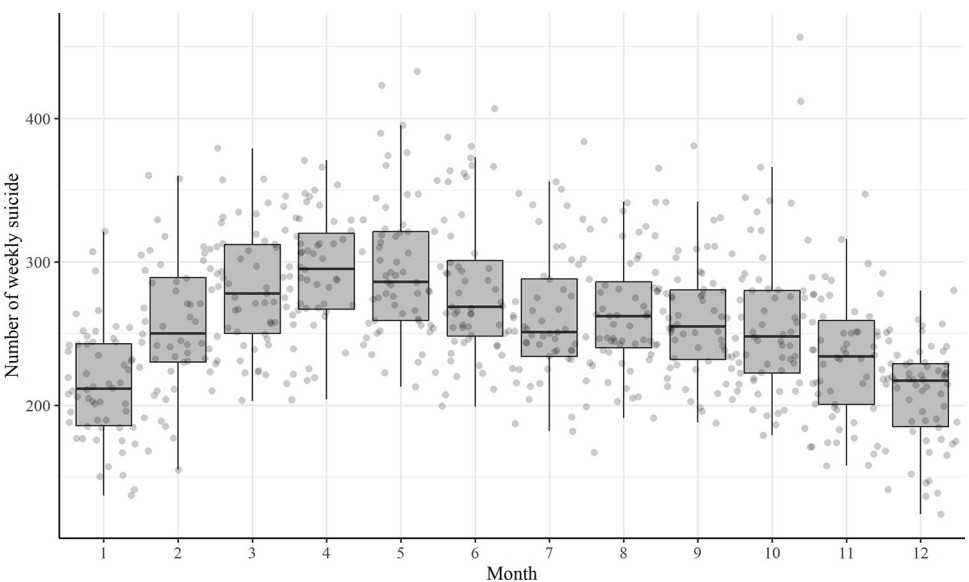

**Fig 2. Monthly variation of weekly numbers of suicides during study period.**

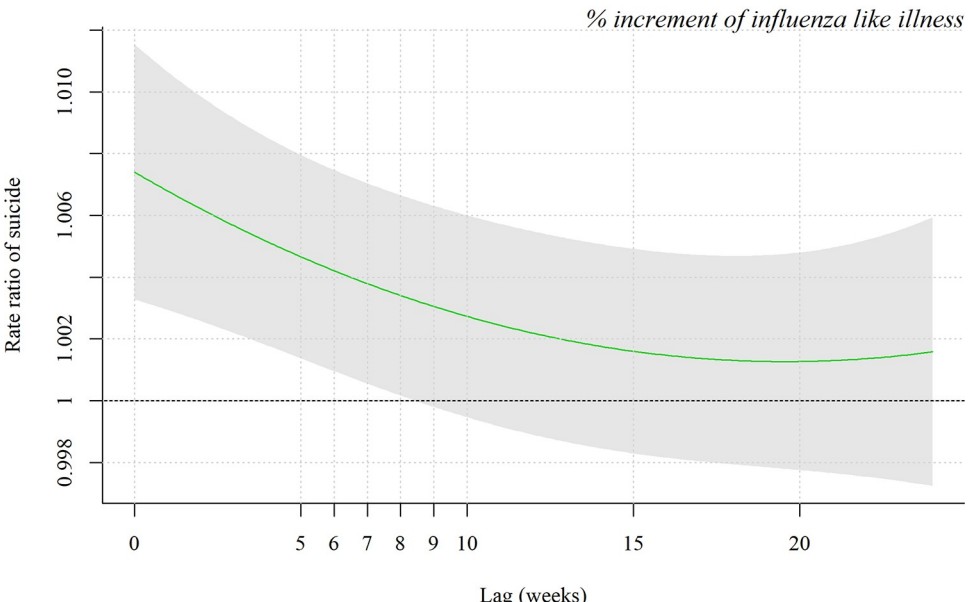

**Fig 3. Association between suicide and influenza according to different lag periods.**

## Discussion

In our study using population data, we identified a significant positive association between ILI and suicide mortality only after the 2009 pandemic, the year in which a novel influenza A (H1N1)pdm09 virus provoked a worldwide pandemic. No meaningful association between these factors was observed before 2009. The association between ILI and suicide mortality increased significantly up to 8 weeks after the onset of ILI symptoms, but 9 weeks after did not.

Corresponding to our results which indicated a significant positive association between ILI number and suicide mortality, one large prospective cohort study [8] using the Danish Civil Registration System showed similar results, supporting our study findings. Among 7,221,578 Danish individuals from 1980 to 2011, hospitalization by viral infection increased the suicide mortality incidence by 1.26 times, compared with control population after adjusting for sex, age, calendar period, cohabitation status, socioeconomic status, and comorbidity. Another prior study [9] of various anti-viral antibody titers in 257 individuals from the U.S. identified a significant relationship between suicidal behavior and influenza B, but not influenza A infection; and a significant association was observed between depression and Influenza A infection. This discrepancy might be attributable to the relatively low statistical power of the previous study. Moreover, suicidal behavior was examined only among people who reported depressive symptoms. Therefore, patients who exhibited suicidal behavior after influenza infection may have not had sufficient time to report depressive symptoms. A large case-control study [2] based on the U.K. Clinical Practice Research Datalink identified an increased risk of depression among people with previous influenza infection, which was only significant 30 days after influenza diagnosis.

Another study of 119 older individuals from the U.S. examined symptoms of depression before and after influenza vaccination and observed no significant change [10]. The possibility of suicidal behavior as an adverse effect of oseltamivir was questioned; however, published

studies, including randomized controlled trials [11, 12] and some observational cohort studies [13], did not find any significant association. A nationally representative case-crossover study [14] from the U.S. analyzed five contemporary influenza seasons (2009–2013) and identified no significant association between oseltamivir use and suicidal behavior. Another longitudinal study [15], including 1.3 million young Danish people, examined the effect of infections and anti-infective agents on suicidal behavior using nationwide registration data, and suggested that although there was a significant positive association between any anti-infective prescription and self-harm, there was no statistically significant relationship between the use of anti-viral agents and suicidal behavior. In our preliminary analysis based on a sampled Korean National Health Insurance (NHI) Cohort, 8,289 patients were diagnosed with influenza during 2010–2013, and 1,835 were exposed to oseltamivir (results not shown); however, the relationship between oseltamivir use and suicidal behavior could not be determined due to the small sample size (three suicide cases in sampled cohort). It has been known that total monthly rate of antibiotic use was highly cross-correlated with the monthly detection rate of influenza virus in South Korea (cross-correlation coefficient 0.47) [16], so it is possible that people who had viral infection might have antibiotic treatment. However, there is a lack of evidence that antibiotics increase the risk of suicidal behavior. Among the antibiotics, fluroquinolone was reported to have some side effects on mental health according to the U.S. Food and Drug Administration; however, suicidal behavior was not included as a side effect.

The biological characteristics of a new virus subtype labeled novel influenza A(H1N1) pdm09 were considerably different from those of the classic (previous) seasonal influenza A (H1N1), which exhibited higher severity in clinical symptoms and strong transmission tendency [14, 17]. The difference might have resulted in a stronger association between ILI and suicidal mortality after 2009. Nevertheless, we could not obtain the weekly data of novel influenza A(H1N1)pdm09 from KCDC, or confirm the magnitude of current association contributed by novel influenza A(H1N1)pdm09.

The etiology of neuropsychiatric symptom development after viral infection remains unknown, and it is unclear whether the virus or corresponding immune response is the causative factor. Several inflammatory cytokines, including interleukin (IL)-2, IL-6, and inferteron-alpha, are assumed to generate the kynurenine neurotoxic metabolite in individuals with suicidal tendencies [18–20] and affect the serotonin system [21], which is a pathway well-known for its influence of depressive disorders [22, 23]: dysregulation of the serotonin system by inflammatory cytokines could partially explain suicidal ideation or behavior among people [24]. Also, a respiratory virus affecting the nose and upper respiratory tract, including sinus infections and allergic rhinitis, has been mentioned as being responsible for central nervous system(CNS) pathologies; viruses such as influenza, coronavirus, and human respiratory syncytial virus are known as a neuropathological virus. Through the neurons which innervate the peripheral organ and olfactory neurons, these viruses enter the CNS, reach the whole brain and cerebrospinal fluid (CSF), and cause demyelination. Simultaneously, primary glial cultures under these conditions have shown the secretion of cytokines in CSF, such as IL-6, IL-15, TNF-α, and CXCL9/10 [25].

In our study, we found lags shorter than 8 weeks with significantly positive association between ILI and suicidal behavior, and the result from Denmark study [8] did not apply any lag, which could imply that their result represent more immediate effect of infection and suicide death. Neuroinflammation can be one of the explanations linking the virus infection and suicidal behavior. Immediate neuroinflammatory changes, including sudden increase in pro-inflammatory cytokines, can exhibit brief symptoms; whereas neuroinflammatory priming, both peripheral and central signaling cells—particularly microglia, exhibit a long-term and derivative change. After stress, primed microloglia produce new "silent" immune machinery,

which causes immune insult and a hyperinflammatory response [26]. Beyond biological mechanisms, the lag between ILI epidemics and suicide include population level analytics, such as window time among exposure and clinical symptoms, or hospitalization.

However, our result indicating seasonal variations in suicidal deaths should be interpreted with other seasonal factors, such as sunlight and pollen exposure. The results from about 43,000 people in Finland showed a significant peak in suicide deaths between May and June, which tend to have the longest sunlight exposure of the year, and the lag period for this association was around 1 month [27]. Contrary to this study, our results showed significant peak of suicidal mortality from March to May, which shows moderate sunlight exposure. Usually, the solstice in South Korea happens in December; therefore, even if we consider the maximum lag to be 8 weeks, there is still a possibility for period overlap due to sunlight exposure. In addition, pollen counts were suspected to be a factor associated with suicide rate [28]; however, a study with data from 42 counties of the continental U.S. did not replicate the significant association [29]. Furthermore, our current model controlled seasonal effects using GAM and time series model, and those adjustments did not attenuate the association between ILI fluctuation and suicide.

The socio-economic difference between times before the 2009 pandemic and post-2009 pandemic should also be considered. In our previous study using national statistics in South Korea, we found an increased suicide rate during the 2008 global financial crisis. We could observe a rise in suicidal mortality, especially among elderly male office workers and managers [30]. Possible interaction may exist between job status, sex, and ILI with suicidal deaths in our data. The novel coronavirus outbreak in 2019 (COVID-19) would differ from novel Influenza A(H1N1)pdm09, in terms of the proportion of affected people, their social influence, and the absence of vaccine and treatment, which may trigger suicidal behavior in more socio-economic context. Further analyses for the adverse mental health outcomes, including suicidal behavior due to these types of respiratory neurotropic viruses, are warranted.

To our knowledge, our study is the first to evaluate the association between ILI infection and suicidal mortality in an Asian population. One report [31] from Hong Kong examined nine cases of neurological complications after H1N1 infection, but did not report the number of suicidal attempts. We utilized nationally representative population data of South Korea, which provided sufficient statistical power to identify significant associations. The time-series analysis enabled us to consider the repeated temporality of both suicidal mortality and ILIs.

However, this study utilized the grouped data, and individual causality could not be interpreted. During modeling, we could not consider important confounders and mediators, including socio-economic and life-style factors or comorbidities. Similarly, there could have been unmeasured confounders such as allergic rhinitis (peaks in airborne pollen), sunlight exposure, and secondary bacterial infections (e.g. sinus infections) after influenza infection, which are also important factors of suicidal behavior, and are associated with seasonal variation.

Various studies showed gender differences in suicidal behavior, and these differences may contribute to biological and social aspects [32]. Regarding viral infection, inflammation and immune response also react differently according to gender [33]. However, we could not perform a gender-specific analysis, since there was no gender-specific information on ILI rate that could be obtained from the original KCDC data. Also, as the information regarding virus genotyping was unavailable, we could not distinguish between H1N1 infection and other flu-like illness.

The dates of ILI diagnosis did not fully represent the true dates of infection. As we used data collected from actual hospitals, the estimates likely did not capture the precise association between the time of the last ILI and suicidal death. Our current results could not confirm

whether novel influenza A(H1N1)pdm09 is the cause of suicidal behavior, due to the lack of weekly information about pathologic subtype. Moreover, age-specific associations could not be identified, as age-specific ILI parameters were unavailable. In addition, as for the potential relationship between influenza vaccination and suicidal behavior, we could not find relevant published research to support the relationship in a population study with a substantial size. Also, we were unable to obtain information on vaccination history in the individual level. Further research to elucidate this relationship is warranted. The U.S. Center for Disease Control and Prevention [34] and the Institute of Medicine Committee to Review Adverse Effects of Vaccines [35] stated that there is no confirmed adverse effect of influenza vaccine on any of neuropsychiatric adverse events, including brain inflammation, encephalopathy, Acute Disseminated Encephalomyelitis, optic neuritis [36], and Guillain-Barre Syndrome [37]. Furthermore, we could not obtain information regarding hospitalization and medication prescription by individuals, since we used national statistics. However, the average hospitalization duration due to ILIs in South Korea is reported to be relatively short (2–3 days) [6], and every hospitalization decision is made during the outpatient visit, including emergency unit visit; therefore, outpatient data was considered to be more sensitive in capturing the influenza cases. As we mentioned earlier, previous studies regarding the relationship between oseltamivir and suicidal behavior were not evident. However, it is possible that other medications for treating flu symptoms could provoke suicidal ideation. For example, ephedrine, a bronchodilator prescribed in severe cold symptoms, is known to have the side effect of suicidal tendencies [38]. In further studies, information about medication after influenza diagnosis should be also considered, in order to clarify which step affects increased suicidal behavior.

## Conclusions

We found a significant positive relationship between ILIs and suicidal mortality after 2009, when a novel influenza A(H1N1)pdm09 virus provoked a worldwide pandemic. The association remained significant until 8 weeks after the ILI development. Further studies with individual-level data are needed to confirm the effects of influenza A infection on suicidal behavior and guide population-based prevention strategies.

## Supporting information

**S1 Box. Korean government's guidelines for reporting "influenza-like illness".**
(DOCX)

## Author Contributions

**Conceptualization:** Sun Jae Jung, Jin-Ha Yoon.

**Data curation:** Sung-Shil Lim, Jin-Ha Yoon.

**Formal analysis:** Jin-Ha Yoon.

**Methodology:** Jin-Ha Yoon.

**Software:** Jin-Ha Yoon.

**Validation:** Sun Jae Jung.

**Visualization:** Jin-Ha Yoon.

**Writing – original draft:** Sun Jae Jung, Jin-Ha Yoon.

**Writing – review & editing:** Sun Jae Jung.

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
