## [Decision Letter · Decision Letter 0]

12 Jun 2020

PONE-D-20-10691

Fluctuations in influenza epidemics and suicide mortality: A time-series regression of 13-year mortality data

PLOS ONE

Dear Dr. Yoon,

Thank you for submitting your manuscript to PLOS ONE. After careful consideration, we feel that it has merit but does not fully meet PLOS ONE’s publication criteria as it currently stands. Therefore, we invite you to submit a revised version of the manuscript that addresses the points raised during the review process.

We look forward to receiving your revised manuscript.

Kind regards,

Mrinmoy Sanyal, PhD

Academic Editor

PLOS ONE

Journal Requirements:

Additional Editor Comments (if provided):

**Please pay attention to the attachment provided by the Reviewer 1.**

Reviewers' comments:

Reviewer's Responses to Questions

**Comments to the Author**

1. Is the manuscript technically sound, and do the data support the conclusions?

Reviewer #1: Partly

Reviewer #2: Yes

2. Has the statistical analysis been performed appropriately and rigorously? 

Reviewer #1: No

Reviewer #2: Yes

3. Have the authors made all data underlying the findings in their manuscript fully available?

Reviewer #1: No

Reviewer #2: Yes

4. Is the manuscript presented in an intelligible fashion and written in standard English?

Reviewer #1: Yes

Reviewer #2: Yes

5. Review Comments to the Author

Reviewer #1: This manuscript investigates an association between suicide mortality and the rate of influenza-like illness (ILI) in Korea. The proposed study is exciting and timely as we experience a virus outbreak. Still, I am afraid that the presentation of statistical methods, data description, the measure of the unit of variables, and analysis is confusing.

I strongly recommend using consistent words for suicide mortality and ILI rate. Throughout the manuscript, I read suicide attempts, influenza symptoms, influenza infections were used instead of suicide mortality, and ILI rate that could distract any reader. Another confusing word was novel A(H1N1)pdm09. Do we have a difference between novel A(H1N1)pdm09, classic novel A(H1N1)pdm09, the new novel A(H1N1)pdm09, A(H1N1)pdm09? I think all are the same, but I would choose one name for the entire text.

I was able to access ILI data, but I couldn't find the suicide rate data.

Please see the attachment.

Reviewer #2: The authors investigated explored the association between influenza infection and suicide rates in a large population using a 57  time-series regression of 13-year mortality data This is an exceptionally timely and commendable work. Several short-comings are well acknowledged in the limitations section. The authors are urged to address the following points

1. Discuss implications and not only list differences between studies that find longer lags and, as as this study, that found shorter lags (here longer than 9 weeks results are not significant). The implications begin with biology- as longer lags imply a sequence of events of cummulation of risk before the maximum predictive association becomes manifest. The authors should discuss the potential of priming of immune cells , including resident immune cells in the brain- process that takes time, and makes the individual more vulnerable to innocuous immune, mechanical and psychological triggers. They could also discuss at a macro level differences between the first Danish study and their studies

2. Avoid the term suicidality. Although there is some terminological confusion, and examples of the use of the term the way the authors have used it so far, for majority of suicide researchers suicidality refers to suicidal ideation not behavior (behavior includes suicide = death by suicide, and non-fatal suicide attempts). The authors would do well to refer to previous studies on attempts as "suicidal behavior" and not suicidality (e.g. citation by Arling), and on studies on death by suicide- as suicide

3. In arguing that the results are not the effect of antivirals the authors write that a Longitudinal study did not find associations with antivirals, but did find associations with antibiotics. It is immortal to report, if data are available - what that percentage of viral infections lead to bacterial complications treated with antibiotics , and what of the percentage of confirmed influenza individuals end up on antibiotics in Korea. If antibiotics may increase risk of suicide - e.g. via reducing the immunomodulatory funciton of the gut microbiota, then it is possible, if large numbers of individuals end up being treated for (real or presumed) respiratory bacterial complications of Influenza, that the antibacterial treatment contributes to results.

4 There is a bit of lack of precision on citations being used, so that when referring the kynurenines- the authors cite nonspecific articles rather that very specific ones ,

a): Sublette ME, Galfalvy HC, Fuchs D, Lapidus M, Grunebaum MF, Oquendo MA, Mann

JJ, Postolache TT. Plasma kynurenine levels are elevated in suicide attempters

with major depressive disorder. Brain Behav Immun. 2011 Aug;25(6):1272-8. doi:

10.1016/j.bbi.2011.05.002. Epub 2011 May 14. PMID: 21605657; PMCID: PMC3468945.

b: Brundin L, Sellgren CM, Lim CK, Grit J, Pålsson E, Landén M, Samuelsson M,

Lundgren K, Brundin P, Fuchs D, Postolache TT, Traskman-Bendz L, Guillemin GJ,

Erhardt S. An enzyme in the kynurenine pathway that governs vulnerability to

suicidal behavior by regulating excitotoxicity and neuroinflammation. Transl

Psychiatry. 2016 Aug 2;6(8):e865. doi: 10.1038/tp.2016.133. PMID: 27483383;

PMCID: PMC5022080.

c: Okusaga O, Duncan E, Langenberg P, Brundin L, Fuchs D, Groer MW, Giegling I,

Stearns-Yoder KA, Hartmann AM, Konte B, Friedl M, Brenner LA, Lowry CA, Rujescu

D, Postolache TT. Combined Toxoplasma gondii seropositivity and high blood

kynurenine--Linked with nonfatal suicidal self-directed violence in patients

with schizophrenia. J Psychiatr Res. 2016 Jan;72:74-81. doi:

10.1016/j.jpsychires.2015.10.002. Epub 2015 Oct 9. PMID: 26594873.

For the immune activation and suicidal behavior- the citations that the author chose are neither the first, nor the most meta-analysis. We suggest replacing them either with the historically first - in the blood, in the CSF, in the brain or with the most recent several meta-analyses.

5 The authors correctly acknowledge overlaps between influenza and other immune and non immune mediated conditions occurring in spring such as pollen exposure and light- but it would be important to cite articles for those, and discuss lags involved in this study relative to the articles on lags of light exposure. The authors may also make the connection between influenza, sinus infections and allergic rhinitis by mentioning the nose/ upper airway to brain pathways for viruses, immune signals and immune cells, further affecting brain structure and function.

6. The differences between before 2009 pandemic and post 2009 pandemic require additional considerations- were there socioeconomic implications or individual psychological stress related to the pandemic contributing to risk. Even if speculative, this component will be very important for COVID-19. In that regard, at this stage the authors would be advised to comment on how the influenza pandemic was different from current COVID 19, and what expectations this article on influenza generates for pandemic of respiratory neurotropic viruses in general

6. PLOS authors have the option to publish the peer review history of their article (what does this mean?). If published, this will include your full peer review and any attached files.

Reviewer #1: No

Reviewer #2: Yes: Teodor T. Postolache, MD

---

## [Author Response · Author response to Decision Letter 0]

24 Jul 2020

We would like to thank the Editor and reviewers for their valuable feedback on our submission to the PLos One. We found your comments to be very helpful in improving the quality of our manuscript. We attached our 'response to reviewer' file

---

## [Decision Letter · Decision Letter 1]

16 Sep 2020

PONE-D-20-10691R1

Fluctuations in influenza epidemics and suicide mortality: A time-series regression of 13-year mortality data in South Korea

PLOS ONE

Dear Dr. Yoon,

Thank you for submitting your manuscript to PLOS ONE. After careful consideration, we feel that it has merit but does not fully meet PLOS ONE’s publication criteria as it currently stands. Therefore, we invite you to submit a revised version of the manuscript that addresses the points raised during the review process.

**Please address The Reviewer #1  comments and return the manuscript as soon as possible. (please see attachment for concerns highlighted in red).**

We look forward to receiving your revised manuscript.

Kind regards,

Mrinmoy Sanyal, PhD

Academic Editor

PLOS ONE

Reviewers' comments:

Reviewer's Responses to Questions

**Comments to the Author**

1. If the authors have adequately addressed your comments raised in a previous round of review and you feel that this manuscript is now acceptable for publication, you may indicate that here to bypass the “Comments to the Author” section, enter your conflict of interest statement in the “Confidential to Editor” section, and submit your "Accept" recommendation.

Reviewer #1: All comments have been addressed

2. Is the manuscript technically sound, and do the data support the conclusions?

Reviewer #1: Yes

3. Has the statistical analysis been performed appropriately and rigorously? 

Reviewer #1: Yes

4. Have the authors made all data underlying the findings in their manuscript fully available?

Reviewer #1: Yes

5. Is the manuscript presented in an intelligible fashion and written in standard English?

Reviewer #1: Yes

6. Review Comments to the Author

Reviewer #1: Thank you for incorporating my comments to revise the manuscript. I found some revisions are missing and highlighted in red color. I am still worried that the figures are blurred. Please see the attachment.

7. PLOS authors have the option to publish the peer review history of their article (what does this mean?). If published, this will include your full peer review and any attached files.

Reviewer #1: No

---

## [Author Response · Author response to Decision Letter 1]

1 Dec 2020

Thank you for your constructive and detail comments

I attached response to reviewer file.

---

## [Editor Report · Decision Letter 2]

14 Dec 2020

Fluctuations in influenza-like illness epidemics and suicide mortality: A time-series regression of 13-year mortality data in South Korea

PONE-D-20-10691R2

Dear Dr. Yoon,

We’re pleased to inform you that your manuscript has been judged scientifically suitable for publication and will be formally accepted for publication once it meets all outstanding technical requirements.

Kind regards,

Mrinmoy Sanyal, PhD

Academic Editor

PLOS ONE

---

## [Editor Report · Acceptance letter]

2 Feb 2021

PONE-D-20-10691R2 

Fluctuations in influenza-like illness epidemics and suicide mortality: A time-series regression of 13-year mortality data in South Korea 

Dear Dr. Yoon:

I'm pleased to inform you that your manuscript has been deemed suitable for publication in PLOS ONE. Congratulations! Your manuscript is now with our production department. 

Kind regards, 

on behalf of

Dr. Mrinmoy Sanyal 

Academic Editor

PLOS ONE